# Analysis of Viral and Host Factors on Immunogenicity of 2018, 2019, and 2020 Southern Hemisphere Seasonal Trivalent Inactivated Influenza Vaccine in Adults in Brazil

**DOI:** 10.3390/v14081692

**Published:** 2022-07-30

**Authors:** Artur Capão, Maria L. Aguiar-Oliveira, Braulia C. Caetano, Thayssa K. Neves, Paola C. Resende, Walquiria A. F. Almeida, Milene D. Miranda, Olindo A.ssis Martins-Filho, David Brown, Marilda M. Siqueira, Cristiana C. Garcia

**Affiliations:** 1Laboratory of Respiratory Viruses and Measles, National Influenza Center (NIC)/World Health Organization (WHO), Oswaldo Cruz Institute, FIOCRUZ, Rio de Janeiro 21040-900, Brazil; artur.capao@ioc.fiocruz.br (A.C.); mlaoliveira@fiocruz.br (M.L.A.-O.); braulia.caetano@ioc.fiocruz.br (B.C.C.); thayssa.silva@ioc.fiocruz.br (T.K.N.); paola@ioc.fiocruz.br (P.C.R.); mmiranda@ioc.fiocruz.br (M.D.M.); david.brown@phe.gov.uk (D.B.); mmsiq@ioc.fiocruz.br (M.M.S.); 2Secretariat of Surveillance in Health (SVS), Ministry of Health (MoH), Brasília-Federal District, Rio de Janeiro 70723-040, Brazil; walquiria.almeida@saude.gov.br; 3Grupo Integrado de Pesquisas em Biomarcadores, René Rachou Institute, FIOCRUZ, Belo Horizonte 30190-002, Brazil; olindo.filho@fiocruz.br; 4UK Health Security Agency, 61 Colindale Avenue, London NW9 5EQ, UK

**Keywords:** influenza, vaccine, immunogenicity, seroprotection, seroconversion, Brazil

## Abstract

Annual vaccination against influenza is the best tool to prevent deaths and hospitalizations. Regular updates of trivalent inactivated influenza vaccines (TIV) are necessary due to high mutation rates in influenza viruses. TIV effectiveness is affected by antigenic mismatches, age, previous immunity, and other host factors. Studying TIV effectiveness annually in different populations is critical. The serological responses to Southern-Hemisphere TIV and circulating influenza strains were evaluated in 2018–2020 among Brazilian volunteers, using hemagglutination inhibition (HI) assays. Post-vaccination titers were corrected to account for pre-vaccination titers. Our population achieved >83% post-vaccination seroprotection levels, whereas seroconversion rates ranged from 10% to 46%. TIV significantly enhanced antibody titers and seroprotection against all prior and contemporary vaccine and circulating strains tested. Strong cross-reactive responses were detected, especially between H1N1 subtypes. A/Singapore/INFIMH-16-0019/2016, included in the 2018 TIV, induced the poorest response. Significant titer and seroprotection reductions were observed 6 and 12 months after vaccination. Age had a slight effect on TIV response, whereas previous vaccination was associated with lower seroconversion rates and titers. Despite this, TIV induced high seroprotection for all strains, in all groups. Regular TIV evaluations, based on regional influenza strain circulation, should be conducted and the factors affecting response studied.

## 1. Introduction

Worldwide, influenza viruses were the major cause of acute respiratory infections before the COVID-19 pandemic and remain a significant threat to public health [1]. These viruses are associated with annual seasonal epidemics, which result in high morbidity and mortality rates and have the potential to cause unpredictable pandemics [1,2]. Annual vaccination is the best available approach to limiting disease burden, especially in certain at-risk groups [3]. Despite increasing global vaccination and surveillance efforts, updated models of influenza-associated lethality from 1999 to 2015 estimate 290,000 to 690,000 deaths each year [4].

Due to high mutation rates and consequent antigenic variation, annual revision and/or the reformulation of influenza vaccines is required to effectively reflect the changing antigenic profile of circulating viruses [2]. However, mismatches between circulating and vaccine strains, as well as variations in each strain’s immunogenicity, have been described in some seasons [5].

A general decline in antibody titers between influenza epidemic seasons has also been reported [6,7,8], corroborating the need for annual vaccination—even when vaccine strains remain unchanged [9,10]. The waning of vaccine-induced immunity over the year is a cause for concern, especially in tropical countries, where the circulation patterns of influenza A and B viruses are spread throughout the year. In Brazil, the peak of influenza A circulation is observed in the autumn and coincides with the annual National Influenza Vaccination Campaign, whereas influenza B circulation is more predominant in the spring and summer (Figure 1) [8,11,12]. Another issue in some reports suggests that repeated annual vaccination can lead to reduced vaccine response [13,14]. This could limit the effectiveness of annual vaccination, which is recommended and distributed free of charge via the Brazilian public healthcare system for at-risk populations, some of which have reduced responses to vaccines [13,14].

Assessments of vaccine immunogenicity and effectiveness on a continual basis are a necessary component of influenza surveillance, providing insights into vaccine effectiveness in different subpopulations, as well as furthering the understanding needed to develop better vaccines, especially in Southern Hemisphere countries where such data are scarce. Higher antibody titers against hemagglutinin (HA), measured by hemagglutination inhibition (HI) assays, correlate with a reduced probability of infection [15]. In this study, we sought to evaluate the serological response to the Southern Hemisphere 2018, 2019, and 2020 Trivalent Influenza Vaccine (TIV) in a Brazilian cohort with HI. The vaccine-induced immunity to circulating influenza strains and the previous year’s vaccine strains were also assessed.

## 2. Materials and Methods

### 2.1. Study Design and Sample Collection

Healthcare workers aged 18 or older and elderly volunteers were enrolled in the study. Volunteers completed a questionnaire for demographic and epidemiological data, including information such as age, vaccination history, influenza infection history, and comorbidities, among others. In 2018, an exploratory recruitment of volunteers over 18 years old who were going to be vaccinated and wanted to join the study was performed. In 2019 and 2020, only volunteers without previous influenza vaccination or who had been vaccinated for the last two consecutive years were included. The studied group was further classified into adult or elderly (under or over 60 years) and into eutrophic or obese (body mass index under 25 or over 30).

In 2018, 2019, and 2020, blood samples were collected prior to vaccination (S1) and 21 days after vaccination (S2). In 2018, an additional blood collection was performed six months after vaccination (S3). 

Vaccination was performed at the Basic Care Health Unit of the National School of Public Health Sergio Arouca (ENSP-Fiocruz, Rio de Janeiro, Brazil) as part of the annual National Influenza Vaccination Campaign in Brazil, using the anti-influenza TIV produced by the Butantan Institute, São Paulo, Brazil.

### 2.2. Virus Isolation

Representative influenza strains circulating in Brazil and isolated at the Laboratory of Respiratory Viruses and Measles—The National Reference Laboratory for the MoH and a National Influenza Center (NIC) from the WHO surveillance network, as well as vaccine viral strains kindly provided by the WHO International Reagent Resource (IRR, Manassas, VA, USA) or by the Francis Crick Institute, London, UK, were cultivated in culture flasks containing confluent monolayers of Madin–Darby Canine Kidney (MDCK) cells in Dulbeco Modified Essential Medium in the presence of 2 μg/mL of TPCK-treated Trypsin (Sigma, St. Louis, MO, USA). Viral titers were determined by hemagglutination assay using guineapig red blood cells. 

### 2.3. Serum Treatment and Hemagglutination Inhibition Assay

Serum samples were treated overnight at 37 °C in 1:4 proportions with Receptor Destroying Enzyme (RDE, Denka Seiken, Tokyo, Japan), followed by heat inactivation for 30 min at 56 °C. Prior to testing, PBS was added for a final serum dilution of 1:10. Ferret or rabbit antisera against viruses—provided by IRR or the Francis Crick Institute—were used as positive controls in the assay and treated with RDE, followed by incubation with guinea pig red blood cells to remove nonspecific hemagglutinins.

For HI assay, treated serum samples were tested in duplicate. Sera were diluted into twofold serial dilutions, followed by a one-hour incubation with each virus at the standard concentration of 4 hemagglutination units per 25 μL. A final one-hour incubation was performed after the addition of 0.75% guinea pig red blood cell suspension. HI titers were determined as the inverse of the highest dilution value in which total inhibition of the hemagglutination was observed. Back-titration was performed for each strain in each assay. Due to the difficulty of some H3N2 strains to hemagglutinate turkey RBCs, guinea pig RBCs were chosen to be used for all strains.

For each year, sera samples were tested against six influenza vaccine viruses. Sera collected in 2018 were tested against the vaccine components A/Michigan/45/2015 (H1N1) (hereafter named H1/MI), A/Singapore/INFIMH-16-0019/2016 (H3N2) (H3/SI), and B/Phuket/3073/2013 (Yamagata-like lineage) (B/PH), as well as three strains that were vaccine components in previous years: A/California/7/2009 (H1N1) (H1/CA), A/Hong Kong/4801/2014 (H3N2) (H3/HK), and B/Brisbane/60/2008 (Victoria) (B/BR). In 2019, sera were tested against vaccine components H1/MI, A/Switzerland/8060/2017 (H3N2) (H3/SW), and B/Colorado/06/2017 (Victoria-like lineage) (B/CO), as well as H1/CA, B/PHm, and H3/SI. In 2020, the respective vaccine components were A/Brisbane/02/2018 (H1N1) (H1/BRI), A/South Australia/34/2019 (H3N2) (H3/SA), and B/Washington/02/2019 (Victoria) (B/WA), as well as H1/MI, B/CO, and H3/SW. Sera collected in 2019 were additionally tested against 6 circulating influenza viruses from different genetic groups, collected throughout our influenza surveillance activities. The selected strains of influenza A H1N1 were A/Rio de Janeiro/1255/2018 (6B.1A1, EPI_ISL_13563916), A/Parana/399/2019 (6B.1A2, EPI_ISL_13724303), and A/Santa Catarina/368/2019 (6B.1A5, EPI_ISL_13724322). From influenza B, we selected one Yamagata strain, B/Santa Catarina/1206/2018 (Y3, EPI_ISL_13724314), and two Victoria strains, B/Parana/69/2019 (V1A.1, EPI_ISL_13723827) and B/Bahia/162/2019 (V1A.1, EPI_ISL_389371), both of which present the double deletion D162 and N163 in the HA gene, also present in the vaccine B/CO strain. 

### 2.4. Patient Follow-Up

Over the course of six months, we established periodic contacts with the volunteers and encouraged self-reporting of influenza-like illness (ILI) symptoms. When ILI symptoms were reported, oro- and nasopharyngeal swabs were immediately collected. After RNA extraction (QIAamp Viral RNA Mini Kit, Qiagen, Hilden, Germany), samples were submitted to influenza (A, B, H1N1, H3N2) detection by one-step reverse transcriptase real-time polymerase chain reaction (RT-PCR), according to the CDC protocol [16]. The influenza HA genes of positive samples were further sequenced using the BigDye Terminator Cycle Sequencing v3.1 kit (Applied Biosystems, Waltham, MA, USA, EUA) and reads obtained by the 3130XL (Applied Biosystems). The read assemblies were performed by the Sequencher v5.1 software (GeneCodes, Ann Arbor, MI, USA), and all sequences were uploaded to the EpiFlu database at GISAID (www.gisaid.org, accessed on 8 July 2022).

### 2.5. Statistical Analysis

HI titers for each sample were calculated as the geometric mean titer (GMT) of the replicates. The GMT values after vaccination were transformed into binary logarithms, a linear regression for each virus was calculated, and then, a correction considering GMT-pre-vaccination was applied (GMT-Post#), according to Beyer et al., 2004 [17], to account for the possible confounding effect of previous immunity (GMT-post-correction = GMT-Post—b × GMT-Pre, where b is the slope of the linear regression). Seroprotection (SP) was defined as titers ≥ 3 (binary log2 of HI titer 40), while the corrected SP post-vaccination values (SP-Post#) were obtained by calculating the corrected threshold for each strain after vaccination. Seroconversion (SC) was defined as increases of at least 4-fold in pre- to post-vaccination GMT.

Accordingly, a comparison of GMT-Post# values was performed using paired or unpaired *t*-tests. After descriptive analysis, putative associations between variables of interest (demographical, clinical, and epidemiological variables) and outcomes (GMT, SC, and SP) were explored. For bivariate analyses, a chi-squared test and Fisher’s exact test were employed. Significance was considered to be *p* < 0.05, and confidence levels of 95% were used. Statistical analyses were performed using the GraphPad Prism version 5.00 software (San Diego, CA, USA).

## 3. Results

### 3.1. Study Population

From the 117 participants enrolled in 2018, the first two blood collections (S1/Pre and S2/Post) were obtained from 113 individuals, and only 95 returned for S3 (Figure 2). The median age was 34 years (range 20–70 years), with a mean of 36.6 years and a median of 34 years, and nine volunteers over the age of 60. The majority of the sample were women (76.9%), and 78.8% of subjects reported receiving at least one vaccine dose before enrollment in the study. These were referred to as the “previously vaccinated group” and received TIV for two or more consecutive years. 

In 2019, 39 participants from the 2018 cohort remained in the study, and 104 new volunteers were recruited. After the second blood collection (S2), our study population consisted of 132 volunteers (Figure 2), with ages ranging from 21 to 86 years, a mean of 44.7 years, and a median of 41 years. A total of 35 patients were older than 60 years (elderly), 72.0% were female, and 64.4% had been vaccinated against influenza in at least the last two consecutive years. In the <60 y and >60 y groups, 57.7% and 85.7% had been previously vaccinated, respectively. A subsample of subjects (16.7%) was obese at grade I or II [18], 73.5% had normal weight (overweight included), and only three volunteers were considered underweight. 

In 2020, our study population consisted of only 30 volunteers because of the COVID-19 pandemic and quarantine regulations (Figure 2). In this last group, with ages ranging from 25 to 60 years, a mean of 38 years, and a median of 36 years, 70.0% were female and 93.3% had been vaccinated against influenza in the last two consecutive years.

### 3.2. TIV Enhances HI Titers and Seroprotection for Current and Previous Years’ Vaccine Strains

For the vaccine components of previous and current years, HI titers before vaccination were overall higher than the seroprotection threshold, which could mask the real effect of vaccination. Indeed, pre-vaccination seroprotection (SP-Pre) rates were above 75.2% for all viruses except for B/CO in 2019, against which only 42.4% of pre-vaccine sera showed antibody levels above the threshold (Table 1). Therefore, we used a correction described by Beyer et al., 2004, by calculating the binary log of GMT post-vaccination titers (GMT-Post) and then applying a linear regression formula considering the pre-vaccination GMT (GMT-Pre). Without Beyer’s correction, the GMT-Pre and GMT-Post demonstrate high pre-vaccination titers with increases after vaccination for all assessed influenza strains (Table 1). When the cohort that was below the seroprotection threshold before vaccination (GMT-Pre < 3) is analyzed separately, the increase in GMT-Post titers is more pronounced (Table 1). In Figure 3A, we show the GMT-Post on the GMT-Pre < 3 population and on the overall population with and without Beyer’s correction. The Beyer-adjusted GMTs (GMT-Post#) were lower than GMT-Post without correction, as shown in Table 1 and Figure 3A.

By comparing SP-Post vaccinations with and without Beyer’s correction, no significant changes were observed (Table 1). After Beyer’s adjustment, we observed that the TIV induced SP-Post# rates ranging from 83% to 100% for the vaccine strains tested (Table 1 and Figure 3B). SP-Post# for H3/SI and B/PH in 2018 were higher than the respective strains in 2019 when they were not the vaccine components; the SP-Post# rate against H3/SI in 2019 was lower than H1/CA and H3/SW, whereas H3/SW SP-Post# was higher than B/CO (Table 1 and Figure 3B). When comparing each year’s vaccine component’s SP-Post# with their previous year’s counterparts, only H3/SW had higher values than their counterpart, H3/SI, in 2019. It is also interesting to note the very similar SC rates between the H1N1s, as well as SP-Post#, and the differences between the influenza Bs of different lineages.

Over the three years, the response to the H1 component was the strongest among the three vaccine constituents (Figure 3C). When comparing each vaccine component across the three years, the response to the 2019 vaccine was the highest (Figure 3C). When comparing vaccine components with the previous year’s vaccine components for each year, we observed that each year’s vaccine component had higher GMT-Post# than their counterparts for H3N2s in 2019 and 2020 and for influenza B in 2018 and 2019, whereas H1N1s present similar GMT-Post# regardless of being a component in the previous year (Table 1). Interestingly, H3/SI in 2018 and B/WA in 2020 induced significantly lower GMT-Post# than their equivalents from the previous year, H3/HK and B/CO, respectively, but with similar SC rates and SP-Post# (Table 1 and Figure 3D).

Individual variations in vaccine response can occur [19]. We observed that only 0.9% (2018) and 6.1% (2019) of subjects seroconverted for all six tested strains, whereas 7.1% (2018) and 13.6% (2019) seroconverted for the three current vaccine components. In both years, volunteers who seroconverted against all strains had never been vaccinated before and were younger than 60 years of age, except for one patient in 2019 who was over 60. Among the 26 subjects who seroconverted for all three vaccine components, only one reported previous vaccination and two were over 60. On the other hand, considering the total of 245 volunteers, 41.6% did not seroconvert for any of the three vaccine components, and 33.5% showed no seroconversion for any strain. In total, 95.1% of these ‘non-responders’ were vaccinated in previous years and 20.7% were older than 60.

### 3.3. TIV Enhances HI Titers and Seroprotection for Matched and Other Contemporary Circulating Strains

Due to the possibility of antigenic mismatch between circulating and vaccine strains, as well as other factors that might modify vaccine immunogenicity, we sought to determine if the vaccine was able to confer protection against representative circulating influenza strains. Using the 2019 subsample, the TIV response was measured by using the serum of 48 volunteers against three circulating influenza A H1N1 strains—more prevalent (41% of detected IV among ILI cases) than H3N2 (18%) in that season (Figure 1)—and three influenza B strains (35,7% of detection in 2019 in Brazil) of different lineages. We observed that the GMT-Post# and SP-Post# against the A/Santa Catarina/368/2019 (6B.1A5) H1N1 strain were similar to the H1/MI. However, the GMT-Post# and SP-Post# against the A/Rio de Janeiro/1255/2018 (6B.1A1) strain were reduced when compared to the vaccine strain H1/MI, whereas the GMT-Post#—but not the SP-Post# against A/Parana399/2019 (6B.1A2)—was significantly higher than the vaccine strain H1/MI (Figure 4A,B). The GMT-Post# against the circulating B strains B/Parana/69/2018 and B/Bahia/162/2019 (Victoria V1A.1) was higher than the vaccine strain B/CO, whereas the GMT-Post# against the circulating B/Yamagata strain, B/Santa Catarina/1206/2018 (Yamagata Y3), was lower than B/CO (Figure 4C). No differences in SP-Post# were detected among the B strains (Figure 4D).

### 3.4. GMT Titers Wane over Time but Remain High through Most Influenza Seasons

In 2018, we observed a significant decrease in antibody titers for all strains after 6 months post-vaccination (S3 GMT-Post#) (Figure 5A). This reduction in GMT was also reflected in seroprotection levels after six months (S3 SP-Post#), which, with the exception of H3/HK, were significantly reduced compared to the after vaccination seroprotection levels (S2 SP-Post#), especially for H3/SI, which presented a higher reduction (Figure 5B).

The uncorrected GMT for H1/MI—the only vaccine strain studied over the 3 years in a smaller cohort that remained in the study (39 individuals from 2018 to 2019 and 13 from 2018 to 2020)—revealed an increase after the first vaccination and a continuous decrease up to the next vaccination, as expected (Figure 5C).

### 3.5. Vaccine Efficacy Appears to Be High, but Does Not Exclude Infection

During the 2018 follow-up, 13 cases of ILI were reported. Among those, only one case was associated with influenza infection, a 37-year-old female without a comorbidity history, who was previously vaccinated. The sample was collected 46 days post-vaccination in 2018, and HA sequencing revealed infection by an A/Michigan/45/2015 (H1N1)-like genotype, 6B.1A1 clade (164T/183P.1). This volunteer did not seroconvert for any of vaccine viruses but was seroprotected for all tested strains except for H1/MI, with S1 GMT (log2) < 1, S2 GMT (log2) 1, and S3 GMT (log2) 6, indicating a seroconversion by natural infection. For the H1/CA strain, she presented titers of 5, 6, and 7.

### 3.6. Previous Influenza Vaccine Influences Induction of Anti-HA Antibodies

Once we observed different patterns of individual response according to influenza vaccine history, the effect of previous immunity to influenza viruses in 2018 and 2019 on vaccine response was explored. We compared the population that had been vaccinated against influenza in the last two consecutive years (VA) to the population that had never been vaccinated against influenza before being enrolled in this study (NVA) as separate groups. GMT-Post# titers were significantly higher in the NVA group for all strains in both years (Figure 6A–C). While the VA variable was associated with a lower SC rate for the three strains (Table 2), no association was found between these variables and the SP-Post# (Figure 6D).

### 3.7. Age Influences Influenza Vaccine Response

Aging induces several alterations in the immune system, reducing the general response against antigens when challenged by natural infection or immunization [3,20]. To investigate the effect of aging on vaccine response, we sought to compare the differences between those over 60 years with the younger population. No differences were detected in the GMT-Post# of vaccine component strains between young or elderly volunteers (Figure 7), while the <60 population had higher GMT-Post# for the previous strains H1/CA and H3/SI in 2019 (data not shown). This variable did not impact SC or SP-Post# for any strain (Table 2 and Table 3). However, the small 2018 sample size for the ≥60 population should be noted, which may result in low statistical power, as well as a possible confounding effect due to the strong correlation between age and previous vaccination status. Using Pearson correlation, inverse correlations between age and H1/MI GMT-Post# (r = −0.22, *p* < 0.05), age and H1/CA (r = −0.23, *p* < 0.01) and age, and H3/SI (r = −0.18, *p* < 0.01) were detected in 2019. The same was detected in linear regressions: H1/MI: r^2^ = 0.05, *p* < 0.05, Y = −0.02 ∗ X + 4.56; H1/CA: r^2^ = 0.05, *p* < 0.01, Y = −0.02 ∗ X + 4.84, and H3/SI: r^2^ = 0.03, *p* < 0.05, Y = −0.02 ∗ X + 2.74.

### 3.8. Other Population Characteristics and Comorbidities Did Not Seem to Impact Vaccine Response

We analyzed the role of obesity, a risk factor for influenza [21,22], in vaccine response, comparing 22 individuals with BMI > 30 (OB) to 97 eutrophic subjects (EG) in 2019. We did not find significant differences in SC, GMT-Post#, or SP-Post# between the different groups (Table 2 and Table 3). Using Pearson correlation and linear regression, no association was found between BMI and GMT-Post# for vaccine components, but a slight inverse Pearson correlation was found for B/PH (r = −0.19, *p* < 0.05), as well as using linear regression (r^2^ = 0.04, *p* < 0.05, Y = −0.06 ∗ X + 3.76).

With the aim of determining if other population characteristics could impact vaccine response, such as sex, recent flu infection, and comorbidities, we performed additional analyses of responses to the 2019 vaccine component strains. Previous analyses for individual comorbidities and other variables did not point to any alterations in vaccine response, but the sample size was very small (data not shown). When grouping all the volunteers with any comorbidity into a single group, we also did not observe any significant impact of this variable on the GMT-Post# or SP-Post#, with only higher SP-Post# for B/PH in the group with comorbidities (Appendix A). Recent flu or flu-like symptoms did not have any impact on the evaluated metrics (Appendix A). Volunteer sex also did not seem to have a significant impact, with only the female sex having higher SP-Post# for H3/SW (Appendix A).

## 4. Discussion

In the present work, we performed a clinical prospective study on a Brazilian cohort of healthy and elderly volunteers to evaluate the 2018, 2019, and 2020 Southern Hemisphere TIV-induced responses against each year’s vaccine strains, past-year vaccine strains, and circulating strains, and we assessed factors that might affect vaccine effectiveness. Our main findings were that: (i) TIVs induced seroprotection against the vaccine component strains in at least 83% of the population studied; (ii) TIV evoked an intense heterologous immunity against previous TIV components and most circulating strains from different antigenic clades/lineages; (iii) the H3N2 component from 2018 TIV H3/SI induced the poorest response; (iv) there is considerable antibody waning after 6 and 12 months, as well as a reduction in SP-Post# levels; (v) repetitive vaccination leads to a less intense, but still effective response; and (vi) indicators of slightly reduced response were found in the elderly.

The SP-Post# values ranged from 83% to 100% after vaccination, slightly higher compared to other studies that evaluated TIV immunogenicity, such as Huang et al. [13], which found values of 39 to 100% for H1N1, 61 to 100% for H3N2, and 60 to 100% for B strains over the course of four influenza seasons, or Xie et al. [23], which found values of 75% to 80% for IV-As and 25% to 35% for IV-Bs, while Mondini et al. [24] and Kieninger et al. [25] observed values closer to ours. It is difficult to draw comparisons between different seasons, vaccine formulations, populations, and studies. This high SP-Post# level may be due to the lower average age of our study population or the high percentage of individuals with preexisting immunity (SP-Pre) [24,26,27]. This could partially explain the very high SP-Post# and the relatively low SC rates, which ranged from 10% to 46% [24,26,27]. It is interesting to note that the B/CO strain, which has a double deletion in its HA that antigenically distances it from previously circulating Victoria strains, was the only strain with GMT-Pre below the seroprotection threshold, which led to the highest seroconversion rates [28]. On the other hand, the same intense response could not be seen for B/WA in 2020, which has a triple deletion, and this may be due to the TIV containing Victoria lineages in consecutive years and the high circulation of influenza B between vaccination campaigns (Figure 1) [29].

We also observed extensive heterologous immunity between the vaccine viruses from the current- and past-year vaccine composition, with some strains that were not in the vaccine inducing a higher response than the current vaccine strains. Despite the differences in GMT-Post# between current- and previous-year strains in most cases, the SP-Post# values were remarkably similar for almost all subtypes. The response for the H1N1s was strikingly similar, which has been reported before [30]. By studying the responses to Brazilian circulating viruses, we observed that vaccination was also able to induce strong protection against several circulating strains, including those that did not belong to the same clade as their vaccine counterpart, with very similar SP-Post# in most cases. The TIV also induced immunity to influenza B circulating strains, two from the B/Victoria lineage, containing the substitutions T210N and V233M in antigenic sites of HA in comparison to B/CO, and, to a lesser extent, one B/Yamagata from the Y3 genetic group. Our results suggest that, even in cases of mismatch, TIVs can retain some degree of effectiveness. However, without a complete vaccination and infection history for each volunteer, the exact cause of this effect could not be determined, and although it is most likely due to boosted cross-reactive antibodies, other factors could also play a role [2,31,32].

Interestingly, the highest seroconversion rates induced by vaccination were observed for the new 2019 vaccine components, B/Colorado/06/2017 and A/Switzerland/8060/2017 (H3N2), but not the new H3N2 component in the 2018 TIV, A/Singapore/INFIMH-16-0019/2016. There was an unusually low response to the A/Singapore/INFIMH-16-0019/2016 strain in 2018, lower than the previous year’s vaccine component A/Hong Kong/4801/2014. A/Singapore/INFIMH-16-0019/2016 also demonstrated a similar response in 2019, when it was not the H3N2 vaccine component. It is also interesting to note that B/CO in 2020 had a higher response than B/Washington/02/2019, the component of the vaccine as discussed above. Although the immunogenicity of the strain itself could explain the observations for A/Singapore/INFIMH-16-0019/2016 (H3N2) in 2018 and B/Washington/02/2019 in 2020, the results seem to be consistent with the original antigenic sin (OAS) theory, discussed below.

Our data showed that, 6 months after vaccination, a significant reduction in the GMT-Post# titers occurred. This also resulted in a significant SP-Post# reduction. With the exception of H3/SI (39%), for which there was a low response, the SP-Post# for vaccine component strains remained over 80% and over 60% for previous-year strains, indicating vaccine-induced protection can last through the influenza season. Our analyses of the volunteers who remained in the study over the years also demonstrate this, with titers continually declining until the next year, as expected, but they also highlight the necessity of annual vaccination even if the vaccine composition remains unchanged.

Previous influenza vaccination seemed to be related to a reduced vaccine response, as demonstrated by the reduced GMT-Post# for most vaccine component strains or past-year vaccine strains. The literature often points to the original antigenic sin theory (OAS) and its derived processes as an explanation, although determining the exact antibody dynamics can be complex [5,14,33,34,35]. Competition between memory and naïve B cells for the same APCs, resulting in either cross-protection or non-neutralizing antibodies, together with an inhibited de novo response, could explain the observations we have made in the previously vaccinated populations [36,37]. Although this effect could also be due to a high level of preexisting antibodies resulting in the neutralization or masking of the antigens. The applied Beyer’s correction was used to compensate for this potential confounder [17,34,38]. Regarding the higher response to the H3/HK strain over the H3/SI strain, it is interesting to note that: (i) the 2017 flu season in Brazil was characterized by high H3N2 circulation (Figure 1); (ii) a high percentage of our study population was vaccinated in 2017; (iii) both strains, due to the culture method, share the T160K mutation [39]; (iv) the NVA group presented a higher response; and (v) despite the three mutations in the HA, studies show H3/SI and H3/HK are antigenically similar [40]. This provides strong indications of backboosting and possible OAS. The higher GMT-Post# for B/CO than B/WA in 2020 can be interpreted in a similar context, with high influenza B circulation during the previous year (Figure 1); a high percentage of volunteers that had been vaccinated in 2019; and studies showing a strong cross-response between these strains’ antigens, even though they present a triple deletion that may distance them from the B/CO strain [41]. However, without a definitive and complete vaccination and infection history for each patient, it is difficult to extrapolate conclusive evidence. Nevertheless, considering the evidence for reduced responses in these groups and the annual character of anti-influenza vaccination campaigns, determining the impact this variable has on vaccine response is of utmost importance. It should be noted that, despite the reduced response, high SP-Post# levels were still induced by the vaccine, and there were no differences between the VA and NVA groups in this regard.

Even if no association was found between age, SC, and SP-Post#, with no differences between the GMT-Post# of the young and elderly as a whole, a Pearson correlation indicated a slight negative correlation between age and GMT-Post#. The literature indicates immunosenescence and inflammaging and their related processes as possible causes for the reduced response in this group, as well as the effects of previous immunity [42,43,44,45]. Some evidence suggests humoral antibody responses may not best represent vaccine responses in this population, especially considering that the cellular response can be altered with advancing age [46,47]. A reduced serological response in this subpopulation would be a serious concern as it is the main risk group with the highest influenza mortality, for which annual vaccination is recommended. The impact of this variable needs to be firmly established so that appropriate strategies can be implemented [48].

Obesity represents a global and growing health concern, with an estimated 13% of Brazilians being obese [49,50]. Obesity has been implicated in increased susceptibility to certain infectious diseases, including influenza, as well as being associated with chronic conditions that can also induce this effect [22,51,52,53,54]. Our results showed only a slight negative correlation between BMI and GMT-Post# for a single strain that was not a vaccine component in that year, indicating obesity had no significant impact on vaccine response in our study population. Although there are conflicting reports on the effect of this variable, a greater decline in antibody levels over time has also been reported in this group, and considering that they are an at-risk group, it is important to monitor this variable [6].

Our volunteer follow-up suggested a high vaccine efficacy. However, it does highlight how population and individual factors can affect vaccine responses. Despite the possibility of infection even in vaccinated individuals, with no mismatch between vaccine strains and circulating strains, the beneficial effect of herd immunity cannot be overlooked, and vaccination remains the best available tool to control influenza.

Our study limitations include the small number of age and body mass index subsamples and the difficulties presented in the recruitment of volunteers in 2020 by the pandemic. Even so, we demonstrated the importance of monitoring TIV immunogenicity on a continuous basis in a country with a distinct pattern of influenza circulation compared to temperate countries, where most studies are performed.

## 5. Conclusions

Overall, we showed that the Southern Hemisphere TIVs from 2018, 2019, and 2020 were effective in inducing protection against each of the component viruses, as well as inducing significant cross-reactions to homologous previous vaccine components and circulating strains. Moreover, our findings indicate that protection lasts through the influenza seasons, but there is a significant reduction after 6 months. Reduced effectiveness in the previously vaccinated and, to a lesser degree, elderly subjects, was observed. Vaccine effectiveness needs to be continuously monitored and new studies need to be conducted to determine the response in the various subpopulations and the mechanisms associated so that more effective vaccines and vaccination strategies can be developed and employed.

## Figures and Tables

**Figure 1 viruses-14-01692-f001:**
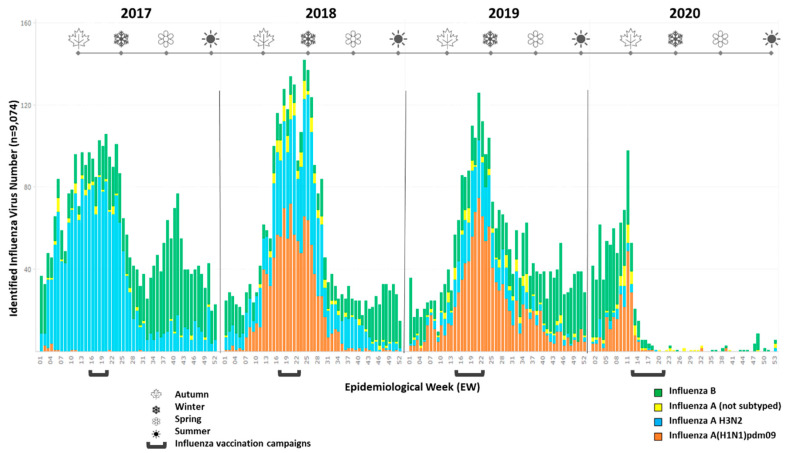
Distribution of influenza viruses among ILI cases from 2018 to 2020 in Brazil. Number of influenza samples identified per epidemiological week and influenza subtype in Brazil using the database of the ILI sentinel system from the Ministry of Health. Influenza B cases are shown in green, influenza A cases not subtyped are in yellow, influenza A H3N2 cases are in blue, and influenza A(H1N1)pdm09 cases are in orange. The beginnings of autumn, winter, spring, and summer are represented by a leaf, snowflake, flower, and sun symbol, respectively, and the period of the National Influenza Vaccination Campaign for each year is marked as a horizontal bracket.

**Figure 2 viruses-14-01692-f002:**
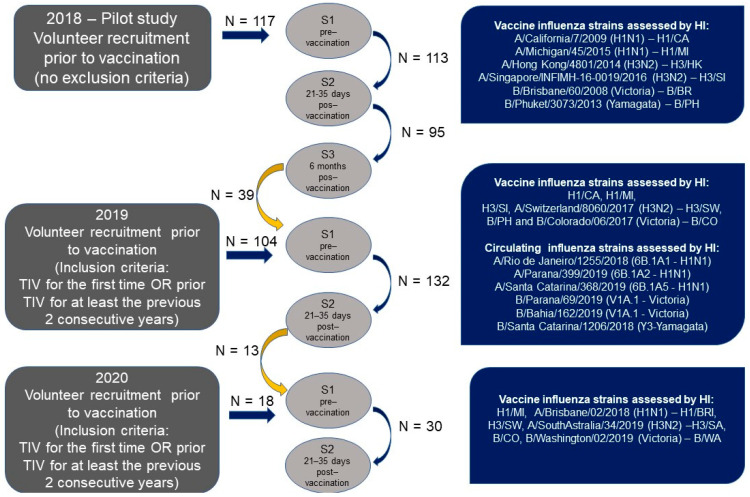
Study design and influenza strains assessed.

**Figure 3 viruses-14-01692-f003:**
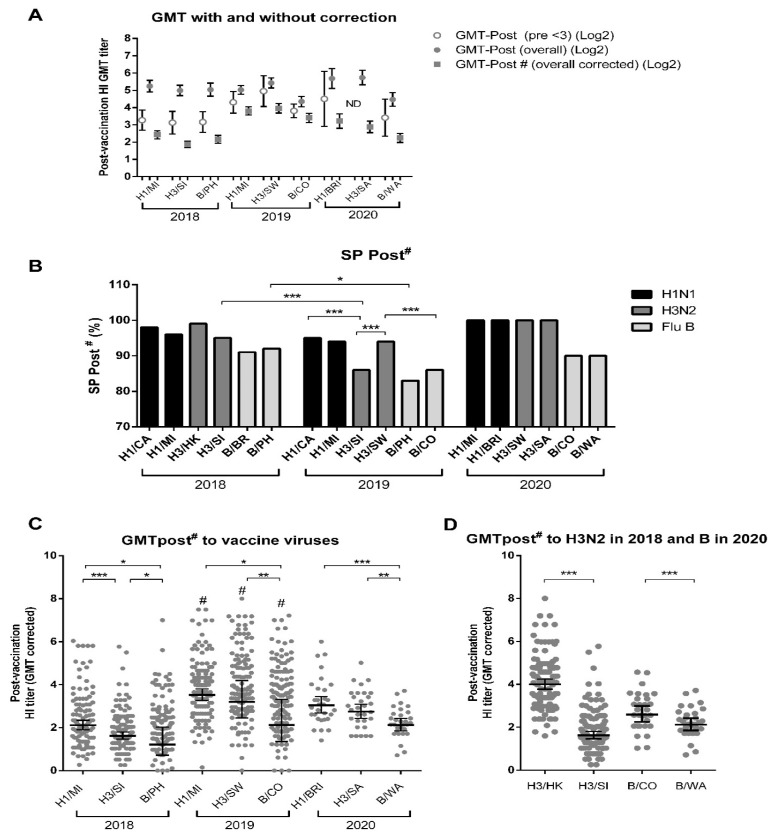
GMT and SP-Post# in the general population for each strain in 2018, 2019, and 2020. (**A**) Comparison between uncorrected binary logarithm post-vaccination GMT titers of the total population (GMT-Post (overall)) for those with pre-vaccination GMT below seroprotection threshold (GMT-Post (Pre < 3) and the total population GMT titers corrected according to Beyer (GMT-Post#). (**B**) Corrected seroprotection levels (SP-Post#). (**C**) Comparison of GMT-Post# of the vaccine component strains of each year. (**D**) Comparison between the GMT-Post# of the vaccine component strain, H3/SI, and the previous year’s strain, H3/HK, in 2018, as well as between the vaccine component strain, B/WA, and the previous year’s strain, B/CO, in 2020. H1/MI = A/Michigan/45/2015 (H1N1); H1/BRI: A/Brisbane/02/2018(H1N1); H3/SI: A/Singapore/INFIMH-16-0019/2016 (H3N2); H3/SW: A/Switzerland/8060/2017 (H3N2); H3/SA: A/South Australia/34/2019(H3N2); H3/HK: A/Hong Kong/4801/2014 (H3N2); B/PH: B/Phuket/3073/2013 (Yamagata); B/CO: B/Colorado/06/2017 (Victoria); B/WA: B/Washington/02/2019(Victoria). Differences between GMT-Post# and between SP-Post# are denoted as * for *p* < 0.05, ** for *p* < 0.01 and *** for *p* < 0.001 in paired *t*-tests and chi-squared tests, respectively. Differences between GMT-Post# of each vaccine component (H1, H3, or B), in different years, are denoted as # for *p* < 0.05 in paired *t*-tests. Sample sizes consisted of 113, 132, and 30 volunteers in 2018, 2019, and 2020, respectively.

**Figure 4 viruses-14-01692-f004:**
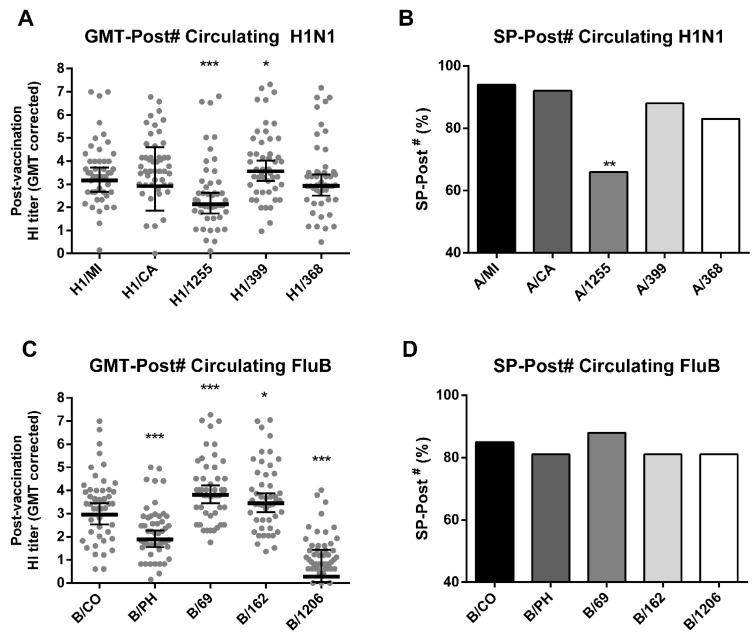
Vaccine response to circulating H1N1 and influenza B strains in 2019. Hemagglutination inhibition geometric mean titers (GMT) corrected post-vaccination (GMT-Post#) for tested circulating H1N1 (**A**) and influenza B (**C**) strains and their vaccine counterparts. Seroprotection post-vaccination with corrected cutoff (SP-Post#) for tested circulating H1N1 (**B**) and influenza B (**D**) strains and their vaccine counterparts. H1/MI = A/Michigan/45/2015 (H1N1); B/PH: B/Phuket/3073/2013 (Yamagata); B/CO: B/Colorado/06/2017 (Victoria); H1/1255: A/Rio de Janeiro/1255/2018; H1/399: A/Parana/399/2019; A/368: A/Santa Catarina/368/2019; B/1206: B/Santa Catarina/1206/2018; B/69: B/Parana/69/2019; B/162: B/Bahia/162/2019. Differences between circulating strains and their respective vaccine strains are denoted as *, **, or *** for *p* < 0.05, *p* < 0.01, or *p* < 0.001 in *t*-tests (**A**,**C**) or chi-squared tests (**B**,**D**). The sample size consisted of 48 volunteers.

**Figure 5 viruses-14-01692-f005:**
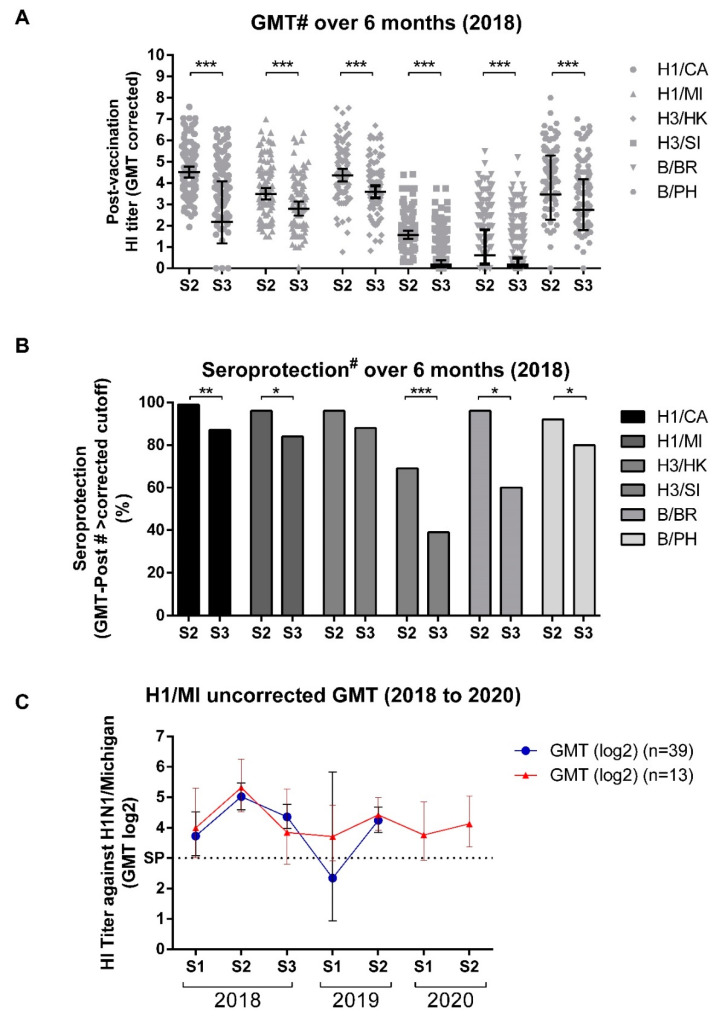
Kinetics of hemagglutination inhibition geometric mean titers (GMT) and seroprotection. (**A**) Beyer’s corrected GMT 21–35 days (S2/GMT#) or 6 months (S3/GMT#) post-vaccination for vaccine strains. (**B**) Seroprotection on S2 or S3 calculated using Beyer’s corrected GMT to adjust the seroprotection cutoff for each assessed virus (Seroprotection#. (**C**) Uncorrected GMT for H1/MI from 2018 to 2019 or 2018 to 2020. H1/CA: A/California/7/2009 (H1N1); H1/MI = A/Michigan/45/2015 (H1N1); H3/HK: A/Hong Kong/4801/2014 (H3N2); H3/SI: A/Singapore/INFIMH-16-0019/2016 (H3N2); B/BR: B/Brisbane/60/2008 (Victoria); B/PH: B/Phuket/3073/2013 (Yamagata). Differences between S2 and S3 are denoted as *, **, or *** for *p* < 0.05, *p* < 0.01, *p* < 0.001 using paired *t*-tests (**A**) or chi-squared tests (**B**). The sample sizes consisted of 95 volunteers for (**A**,**B**) and 13 or 39 volunteers for (**C**).

**Figure 6 viruses-14-01692-f006:**
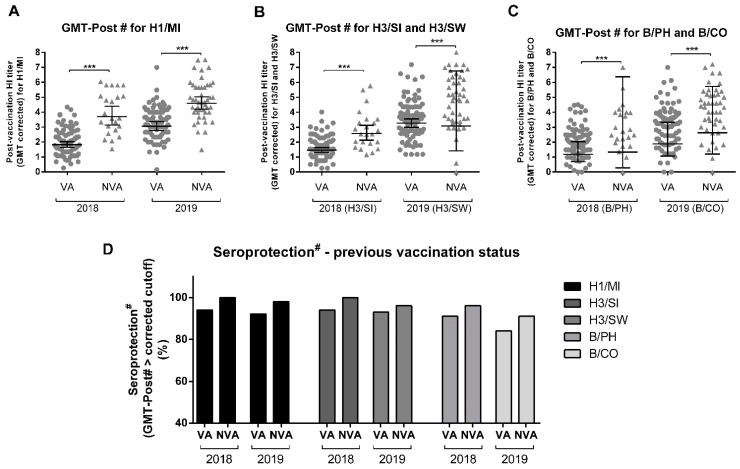
Influence of vaccination history on GMT-Post# and SP-Post#. (**A**) Comparison of post-vaccination GMT corrected according to Beyer (GMT-Post#) and post-vaccination seroprotection with corrected cutoff (SP-Post#) (**D**) in the previously vaccinated (VA) or non-vaccinated (NVA) for A/Michigan (H1N1) in 2018 and 2019. (**B**) Comparison of GMT-Post# and SP-Post# (**D**) in the VA and NVA for A/Singapore (H3N2) in 2018 and A/Switzerland (H3N2) in 2019. (**C**) Comparison of GMT-Post# and SP-Post# (**D**) in the VA and NVA for B/Phuket in 2018 and B/Colorado in 2019. Differences are denoted as *** for *p* < 0.001 in *t*-tests (**A**–**C**) or chi-squared in (**D**). The sample size consisted of 23 NVA and 89 VA in 2018 and 46 NVA and 86 VA in 2019. Circle symbols correspond to VA GMT-Post# and triangles to NVA GMT-Post#, for each strain.

**Figure 7 viruses-14-01692-f007:**
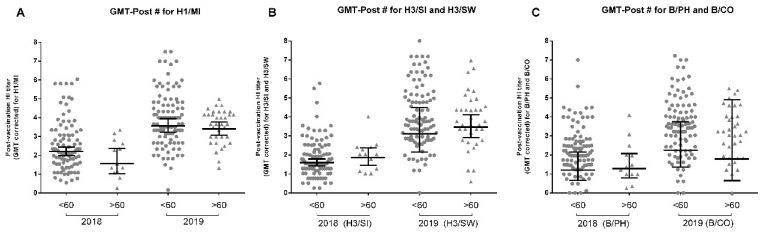
Influence of age on GMT-Post# in 2018 and 2019. (**A**) Comparison of post-vaccination GMT corrected according to Beyer (GMT-Post#) in the young (<60) or elderly (>60) for A/Michigan (H1N1) in 2018 and 2019. (**B**) Comparison of GMT-Post# in the <60 and >60 for A/Singapore (H3N2) in 2018 and A/Switzerland (H3N2) in 2019. (**C**) Comparison of GMT-Post# in the <60 and >60 for B/Phuket in 2018 and B/Colorado in 2019. No differences were detected in *t*-tests. The sample size consisted of 100 < 60 and 13 > 60 in 2018 and 97 < 60 and 35 > 60 in 2019.

**Table 1 viruses-14-01692-t001:** Hemagglutination inhibition geometric mean titers (GMT) pre- and post-vaccination in overall cohorts; GMT-Pre and GMT-Post in subjects below the seroprotection threshold before vaccination (GMT-Pre < 3); seroconversion (SC); seroprotection (SP); GMT-Post with Beyer’s correction; and SP-Post applying Beyer’s correction cutoff for all tested vaccine strains in 2018, 2019, and 2020.

		H1N1			H3N2				Influenza B			
		H1/CA	H1/MI	H1/BRI	H3/HK	H3/SI	H3/SW	H3/SA	B/BR	B/PH	B/CO	B/WA
**2018**	GMT-Pre (Overall) (Log2)	4.8 (1.8)	3.9 (1.7)		4.8 (1.8)	4.2 (1.7)			3.9 (1.4)	3.8 (2.2)		
	GMT-Post (Overall) (Log2)	5.9 (1.8)	5.2 (1.8)		6.2 (1.4)	4.9 (1.6)			4.5 (1.6)	5.0 (2.0)		
	GMT-Pre (Pre<3) (Log2)	1.7 (0.7)	1.6 (0.8)		1.2 (1.1)	1.5 (0.8)			1.3 (0.8)	1.3 (0.9)		
	GMT-Post (Pre<3) (Log2)	3.6 (1.5)	3.3 (1.5)		6.3 (2.1)	3.1 (1.4)			2.5 (1.1)	3.2 (1.8)		
	SC _(fold-increase >4)_ (%)	26	35		26	17			10 **	25 **		
	SP-Pre _(GMT>3)_ (%)	87	75		92	82			75	67		
	SP-Post _(GMT>3)_ (%)	97	92		99	92			83	85		
	GMT-Post # _(corrected)_ (Log2)	2.3 (1.1)	2.4 (1.2)		4.2 (1.3) ***	1.9 (0.9) ***			1.3 (0.8) ***	2.2 (1.2) ***		
	SP-Post # _(GMT-Post#>corrected cutoff)_ (%)	98	96		99	95			91	92		
**2019**	GMT-Pre (Overall) (Log2)	4.2 (1.6)	3.6 (1.5)			3.2 (1.5)	3.6 (1.3)			3.4 (1.7)	2.4 (1.9)	
	GMT-Post (Overall) (Log2)	5.7 (1.6)	5.0 (1.4)			4.4 (1.9)	5.4 (1.7)			4.3 (1.7)	4.3 (1.7)	
	GMT-Pre (Pre<3) (Log2)	1.7 (0.8)	1.7 (0.7)			1.6 (0.7)	1.9 (0.6)			1.8 (0.7)	1.1 (0.8)	
	GMT-Post (Pre<3) (Log2)	4.3 (2.3)	4.3 (1.7)			3.1 (1.9)	4.9 (2.4)			3.4 (1.6)	3.8 (1.7)	
	SC _(fold-increase >4)_ (%)	36	32			25 **	41 **			21 ***	46 ***	
	SP-Pre _(GMT>3)_ (%)	83	77			64	77			60	42	
	SP-Post _(GMT>3)_ (%)	95	95			83	94			80	84	
	GMT-Post # _(corrected)_ (Log2)	3.9 (1.5)	3.8 (1.3)			2.1 (1.5) ***	3.9 (1.6) ***			2.3 (1.4) ***	3.4 (1.5) ***	
	SP-Post # _(GMT-Post#>corrected cutoff)_ (%)	95	94			86 ***	94 ***			83	86	
**2020**	GMT-Pre (Overall) (Log2)		4.5 (1.8)	4.7 (2.0)			4.5 (1.1)	4.8 (1.1)			4.3 (1.5)	3.9 (1.4)
	GMT-Post (Overall) (Log2)		5.1 (1.7)	5.7 (1.5)			5.2 (1.0)	5.7 (1.1)			4.7 (1.4)	4.5 (1.1)
	GMT-Pre (Pre<3) (Log2)		1.5 (1.0)	1.1 (1.3)			-	-			1.7 (0.6)	1.8 (0.9)
	GMT-Post (Pre<3) (Log2)		3.0 (0.0)	4.5 (1.0)			-	-			3.7 (1.5)	3.4 (1.0)
	SC _(fold-increase >4)_ (%)		10	19			10	26			13	13
	SP-Pre _(GMT>3)_ (%)		87	87			100	100			90	80
	SP-Post _(GMT>3)_ (%)		100	100			100	100			97	97
	GMT-Post # _(corrected)_ (Log2)		2.9 (1.1)	3.2 (1.0)			1.9 (0.6) ***	2.9 (0.9) ***			2.8 (0.9) ***	2.2 (0.7) ***
	SP-Post # _(GMT-Post#>corrected cutoff)_ (%)		100	100			100	100			90	90

MI = A/Michigan/45/2015 (H1N1); BRI: A/Brisbane/02/2018 (H1N1); SI: A/Singapore/INFIMH-16-0019/2016 (H3N2); SW: A/Switzerland/8060/2017 (H3N2); SA: A/South Australia/34/2019 (H3N2); PH: B/Phuket/3073/2013 (Yamagata); CO: B/Colorado/06/2017 (Victoria); WA: B/Washington/02/2019(Victoria); GMT = geometric mean titer (presented as mean+/−SD in log2); SC = seroconversion; SP = seroprotection. Statistically significant differences in SC, SP-Post#, and GMT-Post# between current and previous vaccine strains are represented as ** for *p* < 0.01 and *** for *p* < 0.001 in paired *t*-tests or chi-squared tests. # represents values corrected according to Beyer.

**Table 2 viruses-14-01692-t002:** Bivariate analyses evaluating seroconversion for the previously vaccinated (VA) or non-vaccinated (NVA), young (<60) or elderly (>60), and eutrophic (EG) or obese (OB) for the 2019 vaccine component strains.

	A/Michigan/45/2015 (H1N1)	A/Switzerland/8060/2017 (H3N2)	B/Colorado/06/2017 (Victoria)
VA	9 (11%)	23 (27%)	26 (30%)
NVA	33 (72%)	31 (67%)	35 (76%)
Odds-Ratio (CI)	0.05 (0.02–0.12) ***	0.18 (0.08–0.39) ***	0.14 (0.06–0.31) ***
<60	35 (36%)	38 (39%)	48 (49%)
>60	7 (20%)	16 (46%)	13 (37%)
Odds-Ratio (CI)	2.26 (0.89–5.70)	0.89 (0.41–1.90)	1.66 (0.75–3.66)
EG	33 (34%)	47 (19%)	38 (39%)
OB	6 (27%)	10 (46%)	7 (32%)
Odds-Ratio (CI)	0.61 (0.22–1.68)	0.89 (0.35–2.25)	0.73 (0.27–1.94)

Statistical significance is represented as *** for *p* < 0.001.

**Table 3 viruses-14-01692-t003:** GMT-Post# and SP-Post# in the young (<60), elderly (>60), eutrophic (EG), or obese (OB) populations in 2019.

			H1N1		H3N2			Influenza B		
			H1/CA	H1/MI	H3/HK	H3/SI	H3/SW	B/BR	B/PH	B/CO
**2018**	<60	GMT-Post* _(corrected)_ (Log2)	2.4 (1.5)	2.5 (1.3)	4.2 (1.3)	1.8 (1.0)		1.3 (0.8)	2.2 (1.2)	
		SP-Post _(GMT-Post*>corrected cutoff)_ (%)	98	96	99	94		90	93	
	>60	GMT-Post* _(corrected)_ (Log2)	1.9 (0.9)	1.9 (0.9)	4.3 (1.3)	2.0 (0.8)		1.5 (0.7)	1.6 (1.1)	
		SP-Post _(GMT-Post*>corrected cutoff)_ (%)	100	92	100	100		100	85	
**2019**	<60	GMT-Post* _(corrected)_ (Log2)	4.1 (1.6)	3.9 (1.5)		2.2 (1.5)	4.0 (1.6)		2.5 (1.4)	3.5 (1.6)
		SP-Post _(GMT-Post*>corrected cutoff)_ (%)	94	91		89	95		85	88
	>60	GMT-Post* _(corrected)_ (Log2)	3.5 (0.9)	3.5 (0.9)		1.6 (1.3)	3.8 (1.4)		1.9 (1.1)	3.1 (1.4)
		SP-Post _(GMT-Post*>corrected cutoff)_ (%)	97	94		77	91		77	83
**2019**	EG	GMT-Post* _(corrected)_ (Log2)	3.9 (1.6)	3.9 (1.4)		2.1 (1.5)	3.9 (1.6)		2.4 (1.4)	3.5 (1.5)
		SP-Post _(GMT-Post*>corrected cutoff)_ (%)	93	93		87	95		81	90
	OB	GMT-Post* _(corrected)_ (Log2)	4.0 (1.4)	3.8 (1.2)		1.9 (1.5)	4.1 (1.6)		1.8 (1.1)	2.9 (1.6)
		SP-Post _(GMT-Post*>corrected cutoff)_ (%)	100	95		82	86		82	73

## Data Availability

The data presented in this study are available on request from the corresponding author.

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
