# Peer review of "Analysis of Viral and Host Factors on Immunogenicity of 2018, 2019, and 2020 Southern Hemisphere Seasonal Trivalent Inactivated Influenza Vaccine in Adults in Brazil"

_viruses, 2022, doi:10.3390/v14081692_

Round 1

Reviewer 1 Report

The manuscript of Capão et al. present a study of TIV effectiveness in 2018-2020 among Brazilian volunteers and highlights the importance of monitoring TIV immunogenicity.

However I have some questions which can be found below. Addressing these will make the manuscript even better.

1) Did you perform serum microneutralization study? If yes, could you please provide these data?

2) Could you please provide a brief explanation of observed different rates of seroconvection?

3) In the title you clearly state "Impact of previous immunity, body mass index, age ...". But presented data and conclusions do not match the title.

For example, you could perform correlation analysis:

-Summarize your data (figure 6 or other) to get two columns (Age and HI titer)

-Analyse correlation and you can clearly see how age, body mass and other factors correlate with immunogenicity

Author Response

We would like to thank the reviewer for the comments and suggestions, as they certainly contributed to the quality of our work. We have responded to the points and the modifications to the manuscript are detailed below.

1) “Did you perform serum microneutralization study? If yes, could you please provide these data?”

Response: No, we did not perform serum MN, but the HI test is a highly accepted technique to assess immunogenicity and frequently used in vaccine evaluation processes as a correlate of protection measurement [1,2].

2) “Could you please provide a brief explanation of observed different rates of seroconversion?”

Response: Further details were provided on page 14 and Table 1, specifying each viruses seroconversion rate by year. Due to the presence of possible confounding factors in the general cohort and possible different inherent immunogenicity of each strain it is difficult to explain the differences in the seroconversion rates in this general cohorts, it was only possible to observe the effect different variables have, such as previous vaccination. Moreover, a wider discussion about the different seroconversion and the possible explanation via the Original Antigenic Sin (OAS) was presented on pages 26 to 30. Due to the changes imposed by the Beyer correction [3], that was performed after Reviewer 2 suggestion, a lower focus was put on the seroconversion rates in the revised manuscript.

3) “In the title you clearly state "Impact of previous immunity, body mass index, age ...". But presented data and conclusions do not match the title.”

Response: We agree with this comment, and we have made an adjustment on the title. “Analysis of viral and host factors on immunogenicity of 2018, 2019 and 2020 southern-hemisphere seasonal trivalent inactivated influenza vaccine in adults in Brazil.”

4) ”For example, you could perform correlation analysis:

-Summarize your data (figure 6 or other) to get two columns (Age and HI titer)

-Analyze correlation and you can clearly see how age, body mass and other factors correlate with immunogenicity”

Response: We have now performed this analysis for the 2019 cohort, via Pearson correlation and Linear Regression, and adjusted the text accordingly. A slight inverse correlation was found between BMI and a single strain and between age and three strains.

References:

  1. Coudeville L, Bailleux F, Riche B, Megas F, Andre P, Ecochard R. Relationship between haemagglutination-inhibiting antibody titres and clinical protection against influenza: development and application of a bayesian random-effects model. 2010 [cited 2022 Jul 18];10:18. Available from: /pmc/articles/PMC2851702/. DOI:10.1186/1471-2288-10-18.
  2. Antigenic Characterization | CDC [Internet]. [cited 2022 Jul 23]. Available from: https://www.cdc.gov/flu/about/professionals/antigenic.htm
  3. Beyer WEP, Palache AM, Lüchters G, Nauta J, Osterhaus ADME. Seroprotection rate, mean fold increase, seroconversion rate: which parameter adequately expresses seroresponse to influenza vaccination? 2004 Jul [cited 2022 Jul 18];103(1–2):125–32. Available from: https://pubmed.ncbi.nlm.nih.gov/15163500/. DOI:10.1016/J.VIRUSRES.2004.02.024.

Reviewer 2 Report

Impact of previous immunity, body mass index, age and vaccine composition on immunogenicity of 2018, 2019 and 2020 southern-hemisphere seasonal trivalent inactivated influenza vaccine in adults in Brazil.

Artur Capão et al.

General:

The paper describes immune responses to vaccine strains (including previous vaccine strains) and circulation influenza strains over a three year period. The effects of previous vaccination, old age and obesity on sero-responses are also addressed.

Major comments:

It is well known that pre-vaccination titres are a major confounder for mean fold increases and seroconversion rates, where high pre-vaccination titres result in lower MFI and SC. Pre-vaccination titres are highly correlated with postvaccination titres. When analysing MFI and SC a correction can be applied for pre-vaccination titre as described by Beyer et al; the so called Beyer correction. By including pre-vaccination titre as an explanatory variable in the analyses either in an ANCOVA or linear regression model he residual error will be reduced, thereby increasing the statistical power for other explanatory variables. I would recommend that the data are re-analysed using the Beyer correction. See reference to this paper: Beyer, W. E. ., Palache, A. M., Lüchters, G., Nauta, J., & Osterhaus, A. D. M. E. (2004). Seroprotection rate, mean fold increase, seroconversion rate: which parameter adequately expresses seroresponse to influenza vaccination? Virus Research, 103(1–2), 125–132. https://doi.org/10.1016/j.virusres.2004.02.024

Specific comments:

Introduction:

“Another issue arises from some reports, suggesting  that repeated annual vaccination can lead to reduced vaccine response”. Repeated vaccination usually results in higher pre-vaccination titres in the following years, but the GMT’s  may not be different between previously vaccinated- and non-vaccinated subjects. To my mind the GMT is the more important parameter together with the associated seroprotection rate. Higher postvaccination GMTs are a good indication for a longer time of seroprotection and as such seroprotection rates 3 weeks after the last vaccination are a simplification. This has been addressed briefly by the authors.

Methods:

Recent H3N2 strains do not agglutinate well on avian RBC, in the current paper guinea pig RBC were used, was this done specifically to avoid H3N2 agglutination issues? This can be briefly addressed in the discussion.

Results:

The number of elderly subjects (N = 9) in 2018 is low and does not provide sufficient power to investigate age effects. This should be mentioned by the authors. Moreover in the 2019 analysis age and previous vaccinations are highly correlated, making it difficult to make statements on age and or the influence of previous vaccination. This should be mentioned in the text. In 2020 no elderly subjects were included and more than 90% were previously vaccinated.

In figure 3 please add 95% confidence intervals.

In supplementary figures 2, 3 and 4 a paired t-test should have been used rather than an ANOVA.

Author Response

We would like to thank the reviewer for the comments and suggestions, as they certainly contributed to the quality of our work. We have responded to the points and the he modifications to the manuscript are detailed below.

1) “It is well known that pre-vaccination titres are a major confounder for mean fold increases and seroconversion rates, where high pre-vaccination titres result in lower MFI and SC. Pre-vaccination titres are highly correlated with postvaccination titres. When analysing MFI and SC a correction can be applied for pre-vaccination titre as described by Beyer et al; the so called Beyer correction. By including pre-vaccination titre as an explanatory variable in the analyses either in an ANCOVA or linear regression model he residual error will be reduced, thereby increasing the statistical power for other explanatory variables. I would recommend that the data are re-analysed using the Beyer correction. See reference to this paper: Beyer, W. E. ., Palache, A. M., Lüchters, G., Nauta, J., & Osterhaus, A. D. M. E. (2004). Seroprotection rate, mean fold increase, seroconversion rate: which parameter adequately expresses seroresponse to influenza vaccination? Virus Research, 103(1–2), 125–132. https://doi.org/10.1016/j.virusres.2004.02.024 “

Response: We thank the reviewer for the suggestion that was indeed highly relevant. As suggested, we performed the Beyer correction to all GMT data and then reanalyzed the appropriated statistics for each result, including seroprotection rates, which were recalculated based on GMT corrected. 

2) “Another issue arises from some reports, suggesting that repeated annual vaccination can lead to reduced vaccine response”. Repeated vaccination usually results in higher pre-vaccination titres in the following years, but the GMT’s may not be different between previously vaccinated- and non-vaccinated subjects. To my mind the GMT is the more important parameter together with the associated seroprotection rate. Higher postvaccination GMTs are a good indication for a longer time of seroprotection and as such seroprotection rates 3 weeks after the last vaccination are a simplification. This has been addressed briefly by the authors.

Response: We agree and observed a generally higher GMT-Pre in vaccinated individuals, but striking examples of lower GMT-Post# in the vaccinated versus not vaccinated groups. Although the GMT is the most important metric, seroprotection thresholds are frequently used when evaluating influenza vaccine and infection serological responses and allows us to establish a threshold from which to determine protective or not levels of antibodies in the cohort, and thus we felt it was important to relay this metric as well. Due to the Beyers correction being applied, we no longer separately evaluated the seroprotected or not prior to vaccination.

3) Recent H3N2 strains do not agglutinate well on avian RBC, in the current paper guinea pig RBC were used, was this done specifically to avoid H3N2 agglutination issues? This can be briefly addressed in the discussion.

Response: Yes, we chose to use guinea pig RBC for that reason and normalized its use for all viral strains. We now included this information in the methodology section.

4) “The number of elderly subjects (N = 9) in 2018 is low and does not provide sufficient power to investigate age effects. This should be mentioned by the authors. Moreover in the 2019 analysis age and previous vaccinations are highly correlated, making it difficult to make statements on age and or the influence of previous vaccination. This should be mentioned in the text. In 2020 no elderly subjects were included and more than 90% were previously vaccinated.”

Response: In the results section “Age influences influenza vaccine response”, page 23, the low number of elderly individuals in 2018 was noted, we changed the text to make it clearer as well as denoting the possible confounding effect of previous vaccination in this group in 2019.

4) In figure 3 please add 95% confidence intervals.

Response: The color used for the background color did indeed make it very difficult to distinguish from the data points. Regardless, due to the alterations to the manuscript, the MFI data is no longer shown, with similar analyses being presented in Figure 3 C and D.

5) In supplementary figures 2, 3 and 4 a paired t-test should have been used rather than an ANOVA.

Response: After the changes made to the manuscript, due to the Beyers correction, all GMT data were re-analyzed using only paired or un-paired t-tests accordingly.

Round 2

Reviewer 1 Report

Thank you for addressing the comments.